# Identification of Src as a Therapeutic Target in Oesophageal Adenocarcinoma through Functional Genomic and High-Throughput Drug Screening Approaches

**DOI:** 10.3390/cancers14153726

**Published:** 2022-07-30

**Authors:** Niamh H. McCabe, Leanne Stevenson, Enya Scanlon, Rosalie Douglas, Susanna Kennedy, Oliver Keminer, Björn Windshügel, Daniela Zisterer, Richard D. Kennedy, Jaine K. Blayney, Richard C. Turkington

**Affiliations:** 1Patrick G Johnston Centre for Cancer Research, Queen’s University Belfast, Belfast BT9 7AE, UK; nmccabe17@qub.ac.uk (N.H.M.); l.stevenson@qub.ac.uk (L.S.); e.scanlon@qub.ac.uk (E.S.); rdouglas01@qub.ac.uk (R.D.); skennedy2014@qub.ac.uk (S.K.); j.blayney@qub.ac.uk (J.K.B.); 2Fraunhofer Institute for Translational Medicine and Pharmacology ITMP, Discovery Research ScreeningPort, 22525 Hamburg, Germany; oliver.keminer@itmp.fraunhofer.de (O.K.); bjoern.windshuegel@itmp.fraunhofer.de (B.W.); 3Department of Life Sciences and Chemistry, Jacobs University Bremen, 28759 Bremen, Germany; 4Trinity Biomedical Sciences Institute, School of Biochemistry and Immunology, Trinity College Dublin, D08 XW7X Dublin, Ireland; dzistrer@tcd.ie; 5Almac Diagnostics Ltd., Craigavon BT63 5QD, UK; richard.kennedy@almacgroup.com

**Keywords:** oesophageal cancer, chemotherapy, drug resistance, siRNA screen

## Abstract

**Simple Summary:**

Oesophageal adenocarcinoma (OAC) is a leading cause of cancer mortality in the United Kingdom with a 5-year survival rate of approximately 15%. Major contributors to poor outcome are late diagnosis and chemotherapy resistance, and while targeted therapies have benefitted certain cancer settings, they have had limited success in OAC. An understanding of the mechanisms mediating chemotherapy resistance could identify novel targets with the potential to improve standard treatments. This paper aimed to identify mediators of both OAC cancer cell pro-survival signalling and chemotherapy resistance, as well as determine potential small molecular compounds to counteract this. Gene set enrichment analysis of transcriptional data generated a gene-list with significant differential expression between responder and non-responder OAC patients. Gene functionality assessment using siRNA screening showed that targeting SRC had an anti-tumour effect in OAC cells with the potential to enhance chemotherapy treatment. In parallel to this, a compound screen showed the Src inihibitor dasatinib sensitised OAC cells to chemotherapy. Together, these findings suggest targeting SRC as a novel therapeutic strategy in OAC.

**Abstract:**

Drug resistance limits the effectiveness of oesophageal adenocarcinoma (OAC) chemotherapies, leading to a poor prognosis for this disease. Elucidation of the underlying resistance mechanisms is key to enabling the identification of more effective treatments. This study, therefore, aims to identify novel therapeutic and/or chemotherapy sensitising drug targets in OAC. Transcriptional data from a cohort of 273 pre-treatment OAC biopsies, from patients who received neoadjuvant chemotherapy followed by surgical resection, were analysed using gene set enrichment analysis (GSEA) to determine differential gene expression between responding and non-responding OAC tumours. From this, 80 genes were selected for high-throughput siRNA screening in OAC cell lines with or without standard chemotherapy treatment. In parallel, cell viability assays were performed using a panel of FDA-approved drugs and combination index (CI) values were calculated to evaluate drug synergy with standard chemotherapy. Mechanisms of synergy were investigated using western blot, propidium iodide flow cytometry, and proliferation assays. Taken together, the screens identified that targeting Src, using either siRNA or the small molecule inhibitor dasatinib, enhanced the efficacy of chemotherapy in OAC cells. Further in vitro functional analysis confirmed Src inhibition to be synergistic with standard OAC chemotherapies, 5-fluorouracil (5-FU), and cisplatin (CDDP). In conclusion, a compound screen together with a functional genomic approach identified Src as a potential chemosensitising target in OAC, which could be assessed in a clinical study for poor prognosis OAC patients.

## 1. Introduction

Incidence rates of OAC have progressively increased by six-fold since the 1970s, with the United Kingdom reported to have one of the highest global incidence rates together with the poorest survival rates [1]. Standard therapy for resectable OAC includes either neoadjuvant chemoradiotherapy or chemotherapy followed by surgical resection in patients with localized disease. These patients have a low pathological response rate of 15% and 5-year survival rate of only 45% [2,3]. However, upon relapse, OAC is an aggressive disease with most patients presenting with distant metastatic disease and a low 5-year survival of 13% [4,5]. Consequently, new treatment approaches are needed to improve survival.

Resistance to chemotherapy is a major problem in the treatment of cancer, hence the clinical urgency to identify key molecular targets. The expression or mutational status of these targets may constitute novel predictive biomarkers of drug response or resistance. In addition, these molecular determinants of drug resistance may provide novel therapeutic strategies for enhancing the clinical effectiveness of current chemotherapeutics.

The aim of this study was to apply high-throughput technologies such as siRNA and compound screens to identify molecular determinants of drug resistance in OAC. We discovered pathways and genes differentially regulated between chemotherapy non-responders and responders by applying gene set enrichment analysis (GSEA) to the transcriptional data from 273 pre-treatment OAC biopsies [6]. A functional genomic approach using siRNA was then applied to evaluate if targeting these genes enhanced OAC cell line sensitivity to the standard chemotherapeutics, 5-fluorouracil (5-FU) and cisplatin (CDDP). A high-throughput screen of FDA-approved drugs was also applied to test the efficacy of compounds to sensitize to chemotherapy in OAC cells. This approach identified inhibition of Src as a chemosensitization strategy in OAC.

## 2. Materials and Methods

### 2.1. Cell Culture

OE33 (Cat. No. 96070808), FLO-1 (Cat. No. 11012001), SKGT4 (Cat. No. 11012007), and OE19 (Cat. No. 96071721) OAC cell lines were obtained from Public Health England. MFD-1 cells were a kind gift from Professor Tim Underwood, University of Southhampton [7]. OE33, SKGT4, and OE19 cells were maintained in RPMI growth media, while FLO-1 and MFD-1 cells were maintained in DMEM growth media (ThermoFisher, Loughborough, UK, Cat. No 11875093 and 11965092). Growth media was supplemented with 10% foetal bovine serum (FBS) (ThermoFisher Cat. No 10500064), 1% sodium pyruvate (ThermoFisher Cat. No 11360070), and 1% penicillin/streptomycin (ThermoFisher Cat. No 15140122) at 37 °C in a humidified atmosphere containing 5% CO_2_. All cell lines were routinely examined for mycoplasma contamination using PlasmoTest^TM^ (Invivogen, Toulouse, France, Cat. No. rep-pt1).

### 2.2. Cell Viability Analysis

Cell viability was determined at 48 h or 72 h post-treatment using MTT 3-(4,5-dimethylthiazol-2-yl)-2,5- diphenyltetrazolium bromide (Sigma-Aldrich, Gillingham, England, Cat. No M2003) or CellTiter-Glo^®^ assays (Promega, Southhampton, UK, Cat. No. G7570). Briefly, exponentially growing cells were seeded into 96-well plates and left to incubate overnight in a total volume of 100 µL/well (5% CO_2_ at 37 °C). Cells were treated 24 h post-seeding to obtain a total volume of 200 µL/well and viability was measured at 72 h post-treatment. For the MTT assay, 20 µL of 5 mg/mL MTT was added to each well for 3 h (5% CO_2_ at 37 °C). The culture media was then aspirated, and the remaining formazan crystals were dissolved with 70 µL of DMSO. Cell survival was determined by reading the absorbance of each well at 570 nm using a Synergy II microplate reader and Gen5 software (BioTek). For the CellTiterGlo^®^ assay, viability was assessed according to the manufacturer’s instructions. Dose–response curves were generated using Prism 8.0 software (GraphPad Software Inc., version 8.0, San Diego, CA, USA) to calculate the ~IC_30(72 h)_ and ~IC_50(72 h)_ drug doses.

### 2.3. Combination Index Analysis

Effective drug combinations were assessed for synergy using the Chou and Talalay method generating a combination index (CI) [8,9]. This was performed using Compusyn version 1 software, where CI values <1, 1, and >1 indicate synergy, additivity, and antagonism, respectively. For synergistic interactions, CI values between 0.8 and 0.9 indicate weak synergy, between 0.4 and 0.8 indicate moderate synergy, and <0.4 indicate strong synergy.

### 2.4. Cell Proliferation Rates

At 24, 48, and 72 h post-treatment with ~IC_30(72 h)_ doses of Src inhibitor, chemotherapy, or a combination, viable cells were assessed using 0.4% trypan blue (NanoEntek, Seoul, South Korea, EBT-001) exclusion [10]. Viable cells were counted using EVE^TM^ counting slides (NanoEntek, Cat# EVS-050) with the Countess II automated cell counter (ThermoFisher). All samples and time-points were normalized to the 24 h control sample.

### 2.5. Flow Cytometry

The DNA content of harvested cells was evaluated after propidium iodide (PI) staining of cells using the BD sampler plus FACS calibur system. PI cell cycle analysis was used to quantify the sub-G1/G0 population. Cells were harvested and analyzed according to the manufacturer’s instructions (BD Biosciences, San Jose CA, USA) following 72 h treatment.

### 2.6. Western Blotting

Western blot analysis was carried out using antibodies targeting phospho-(Y416)-Src Rabbit pAb (Cell Signaling Technology, Danvers, MA, USA, Cat# 2101, RRID:AB_331697), Src (L4A1) Mouse mAb (Cell Signaling Technology Cat# 2110, RRID:AB_10691385), and PARP (46D11) Rabbit mAb (Cell Signaling Technology Cat# 9532, RRID:AB_659884). Mouse monoclonal antibodies were used in conjunction with anti-mouse IgG, HRP-linked Antibody (Cell Signaling Technology Cat# 7076, RRID:AB_330924). Rabbit polyclonal antibodies were used in conjunction with anti-rabbit IgG, HRP-linked Antibody (Cell Signaling Technology Cat# 7074, RRID:AB_2099233). Equal loading was assessed using vinculin (Abcam Cat# ab129002, RRID:AB_11144129) or β-actin (Cell Signaling Technology Cat# 3700, RRID:AB_2242334) targeting antibodies.

### 2.7. High-Throughput Compound Screening

OE33 and FLO-1 cells were seeded at 500 cells/well in 384-well plates (Greiner Bio-One GmbH, Frickenhausen, Germany) and incubated overnight. After 24 h incubation, OE33 cells were either treated with 0.1% DMSO control, 4 µM cisplatin (2 X positive control), and 1 µM of library compound, or co-treated with a mixture of ~IC_50_ 2 µM of CDDP and 1 µM of a library compound for 72 h. After 24 h incubation, FLO-1 cells were either treated with 0.1% DMSO, 10 µM cisplatin (2 X positive control), and 1 µM of library compound, or co-treated with a mixture of IC_50_ 5 µM of cisplatin and 1 µM of a library compound for 72 h. Compound transfer was carried out using Echo^®^ 550 Liquid Handler (Labcyte, Sunnyvale, CA, USA) from 10 mM stock plates of the SCREENWELL^®^ FDA approved drug library V1 (ENZO Life Sciences GmbH, Körrach, Germany). Following 96 h total incubation, cell lysates were assayed for ATP activity, using CellTiter-Glo^®^ (Promega, Cat. No G7570) according to the manufacturer’s instructions. Output was read on the EnVision multilabel plate reader (PerkinElmer, Rodgau, Germany). The effect on cell viability was measured relative to the DMSO control and then compared to the 2X dose of CDDP only (positive control). Test compounds that caused 80% lower viability levels than the DMSO control were deemed as potential ‘hits’ that would be considered for further evaluation.

A further quality control step was required to show that compounds did not interfere with inherent ATP activity and subsequently alter the results of CellTiter-Glo^®^. Therefore, all compounds in the library were tested in cell-free growth medium containing 1 µM ATP. This was compared to an ATP titration with 0.1% DMSO and measured by CellTiter-Glo^®^ following 24 h incubation.

Quality and robustness of the screen were evaluated by calculation of the Z’ factor using the formula Z’ = 1 − 3 (δp + δn)/(|µp − µn|), where µ is the mean and δ is the standard deviation of positive and negative controls [11]. Calculated on a plate-by-plate basis, the Z’ factor allows to determine whether an assay resulted in responses large enough to justify further investigation.

### 2.8. Bioinformatic Analysis of OAC Clinical Dataset

A dataset of 273 pre-treatment biopsies from OAC patients undergoing platinum-based neoadjuvant chemotherapy and surgical resection has been previously described [6]. This included patient demographics, clinical characteristics, chemotherapy regimen, and measurement of response. Gene set enrichment analysis (GSEA) is a computational analysis platform developed by the Broad Institute to assess where prior defined gene sets within the Molecular Signature Data Base (MSigDB) curated gene-signature libraries show statistically significant, concordant differences between two biological phenotypes being compared [12,13,14]. Our analysis carried out GSEA on our dataset of n = 273 samples, where samples were divided into responder (n = 26) and non-responder (n = 247) phenotypes determined by tumour regression grade at surgical resection, following platinum-based neoadjuvant chemotherapy [6]. We selected the MSigDB v6.0 C2 library inclusive of Kyoto Encyclopaedia of Genes and Genomes (KEGG) and Reactome curated signatures as well as the C5 library representing Gene Ontology (GO) curated signatures to determine significantly enriched pathways representing biological function and disease within the phenotypes analysed. A normalised gene-expression matrix of our samples, Almac Xcel Array Chip file, and phenotype annotation files were loaded into GSEA, with analysis parameters set to C2 and C5 gene-set databases, selected to run on 1000 permutations using a weighted enrichment statistic, with minimum and maximum gene-set sizes set to 15 and 500, respectively, where analysis would exclude gene sets above or below the threshold. Using these pre-defined parameters, GSEA then identified significantly positively or negatively enriched pathways per phenotype by two steps (i) a nominal *p*-value of <0.01 and a (ii) a false discovery rate (FDR) 0.25, while highlighting specific genes within an enriched gene-set, which drive the phenotypic enrichment.

### 2.9. Generation of CDDP-Resistant Cell Line

A CDDP-resistant (CDDP-R) OE33 cell line was generated in our laboratory. The parental OE33 cell line was continuously exposed to increasing concentrations of CDDP over 6 months. Following each passage, the cells were initially drugged with a ~IC_10(72 h)_ concentration of CDDP. The CDDP dose was then gradually increased as the cells’ sensitivity to CDDP decreased. Sensitivity was measured using MTT, generating dose–response curves using Prism 8.0 software (GraphPad Software Inc.) to confirm that these sub-lines were more resistant to CDDP than the original parental counterparts.

### 2.10. siRNA Transfection

All siRNAs were purchased from Qiagen, Manchester, UK, (Cat. No. 1027411). All Stars negative control (NC) was used as a non-targeting scrambled control siRNA (Qiagen, Cat. No. SI03650318) and All Stars Death (Qiagen, Cat. No. SI04381048) was used as a lethal positive control siRNA. OAC cells were reverse-transfected using siRNA in Opti-MEM^TM^ (Thermo Fisher, Cat. No. 11524456) and HiPerfect transfection reagent (Qiagen, Cat. No. 301707) to a final concentration of 10 nM. Cells were treated with a drug at 24 h post transfection and knock-down of the targeted gene was assessed at 72 h post-transfection.

### 2.11. Quantitative PCR (Q-PCR)

Total RNA was isolated using RNA STAT-60 reagent according to the manufacturer’s instructions (AMS Biotechnology, Abington, UK, Cat. No. CS-502). Reverse transcription was carried out using Transcriptor first strand cDNA synthesis kit (Roche Diagnostics, Basel, Switzerland, Cat. No. 04896866001) according to the manufacturer’s instructions. Reverse transcription-PCR (RT-PCR) amplification was carried out in a final volume of 10 μL containing 5 μL of LightCycler 480 Probes Master (Roche Diagnostics, Cat. No. 04707494001) and 1 μL of RealTime ready Assay (Roche Diagnostics, Cat. No. 05532957001), 1.5 μL of PCR grade H2O (Roche Diagnostics), and 2.5 μL of cDNA using a Light Cycler^®^ 480 II (Roche Diagnostics) according to the manufacturer’s protocols.

### 2.12. Statistical Analysis of In Vitro Experimental Replicates

All *t*-tests were calculated using the GraphPad software (Prism 8.0) and were unpaired, two-tailed tests using 95% confidence intervals.

## 3. Results

### 3.1. GSEA Identified Enriched Novel Pathways and Genes in Responder and Non-Responder Oac Tumours

In this study, although the sample size was large, the number of responders was too small to allow any pathways to be enriched following application of the FDR 0.25 (26 responders out of 273, <10%), so analysis was performed on the enriched pathways assessed by nominal *p*-value at a cut-off of *p* <0.01. The GSEA program tested the two groups (responders and non-responders) on whether there were significant differences between them in terms of their gene expression. Leading edge analysis was included within the gene-set enrichment analysis to determine driver genes within the significantly dysregulated biological pathways identified. This allowed us to determine which genes were driving phenotypic biology and whether they were up or down regulated in our phenotype of interest. A total of 43 pathways were significantly enriched between responders and non-responders (Table 1). When assessing the non-responders, there was a total of 18 enriched pathways: 3 enriched in C2 resulting in 42 enriched genes and 15 enriched in C5 with 555 enriched genes. In the responders, there were a total of 25 enriched pathways: 8 enriched in C2 with 272 genes and 17 enriched in C5 with 271 genes enriched. Examining the pathways enriched in non-responders, there was a high frequency of pathways related to mitochondrial and metabolic dysfunction.

The pathways enriched in the responder group show a focus on response to stimuli, with most pathways corresponding to reactions to specific stimuli such as prostaglandins, producing immune or inflammatory responses.

As some overlap of genes existed within each group, the GSEA resulted in 784 enriched genes (Appendix A), from which 80 candidate genes from the top-ranked pathways were selected for siRNA screening. The following criteria were applied for gene selection: (1) Biological relevance to OAC. For this, a literature search was performed for each gene using the PubMed database. (2) Overexpression in oesophago-gastric cancer was identified using ProteinAtlas and UniProt databases. (3) Interactants for each gene were also taken into consideration and pathway signalling was determined using the KEGG pathway database. (4) Potentially targetable candidates were considered preferential such as proteins that were membrane bound, possessed enzymatic activity, or had an inhibitor available.

By applying these criteria, our starting list of 784 genes was filtered down to 80 candidate genes for siRNA screening and can be further sub-divided by Hanahan and Weinberg’s Hallmarks of Cancer (Figure 1) [15].

### 3.2. Functional Assessment of Selected Genes in OAC Cells

The 80 selected target genes were investigated for their functional effect in OAC, either alone or in terms of mediating chemotherapy resistance. For this, OE33 cell viability was measured by MTT following 24 h siRNA-mediated knockdown ± 48 h treatment with ~IC_30(72 h)_ CDDP or 5-FU treatment (Appendix A). This primary siRNA screen identified 24 genes that caused a significant decrease in viability with siRNA alone (Appendix A). These 25 genes were then taken forward to a secondary siRNA screen for further validation. Here, three siRNA sequences per target, inclusive of the primary screen siRNA, were assessed as before. A positive hit was determined by a minimum of two out of three siRNA sequences significantly reducing OE33 viability either alone or in combination with chemotherapy. This identified BACH1, F2RL2, FGF10, NNT, PKM2, SRC, and STAR genes (Appendix A).

To rule out OE33-specific effects, these seven target genes were taken forward to a tertiary siRNA screen. The most potent siRNA from the secondary screen was used in a panel of OAC cell lines consisting of OE33, FLO-1, SKGT-4, and MFD-1. In agreement with the secondary screen, all seven target genes significantly (*p* < 0.05) reduced OE33 viability with either knockdown alone or in combination with CDDP (Figure 2a). In FLO-1 cells, knockdown of BACH1, F2RL2, and STAR alone or in combination with CDDP significantly (*p* < 0.05) decreased cell viability (Figure 2b). In SKGT4 cells, all seven targets, either alone or combined with CDDP or 5-FU, significantly (*p* < 0.05) decreased viability (Figure 2c). In MFD-1 cells, knockdown of BACH1, FGF10, NNT, SRC, and STAR alone or in combination with CDDP significantly (*p* < 0.05) decreased cell viability (Figure 2d). These results confirmed that anti-tumour and/or chemotherapy enhancing effects of target knockdown were not limited to a single cell line model and were effective across the cell line panel.

### 3.3. Identification of CDDP Enhancing Compounds in OAC Cell Line Models

As platinum-based chemotherapies are the main active DNA damaging agents used to treat OAC, CDDP sensitization was the focus of this study. OE33 and FLO-1 cells were treated with 775 compounds from the SCREEN-WELL^®^ V1 FDA-approved drug library. All compounds were tested at 1 µM either alone or in combination with ~IC_50(72 h)_ doses of CDDP. Following 72 h treatment, cell viability was measured using the Cell Titer Glo^®^ assay. A total of 97 compounds, either alone or in combination with CDDP, were found to decrease cell viability by ≥80% in OE33 cells and/or FLO-1 cells (Figure 2e, Appendix A). Of note, the antineoplastic drug dasatinib has been shown to be a potent inhibitor of Src and Src family kinases (SFKs) [16].

### 3.4. Validation of SRC Knockdown and Phospho-Src Y416 Inhibition

Of the seven potential OAC targets identified from this siRNA screen, Src is the most clinically evolved with inhibitors trialed in multiple cancer types [17,18]. It is also a key nodal signalling molecule for other targets identified in the screen (FGF10 and PKM2) [19,20]. Furthermore, the Src inhibitor dasatinib was identified as a significant compound in the repurposing screen. Therefore, Src was selected for further functional validation as a potential chemosensitizing target in OAC.

Initially, basal levels of total- and phospho-Src tyrosine 416 (Y416) were assessed across a panel of OAC cell lines: OE33, FLO-1, SKGT4, OE19, and MFD-1 (Figure 3a). This showed that the OE19 and OE33 cells had the highest levels of both total Src and phospho-Src Y416, with moderate levels observed in the FLO-1 cells and the lowest levels detected in the SKGT4 and MFD-1 cells. OAC cell line sensitivity to the Src inhibitors dasatinib and saracatinib was assessed using MTT viability assays to calculate ~IC_30(72 h)_ doses following 72 h treatment (Appendix A). However, neither levels of phospho-Src Y416 nor total-SRC were indicative of sensitivity to these Src inhibitors as single agents. Knockdown of Src in the OE33, FLO-1, SKGT4, and MFD-1 cells following 72 h siRNA transfection was confirmed at the protein level by Western blot analysis (Figure 3b). Knockdown of SRC was also confirmed at the gene expression level by Q-PCR (Figure 3c). Of note, the OE19 cells had a low siRNA transfection efficiency and this cell line was excluded from further studies.

For Src inhibitor investigations, three OAC cell lines were selected based on high (OE33), medium (FLO-1), and low levels (MFD-1) of total-Src and phospho-Src Y416 (Figure 3a). Inhibition of phospho-Src Y416 following 24 h treatment with dasatinib or saracatinib was confirmed by Western blot. In the OE33 and FLO-1 cells, basal phospho-Src Y416 was inhibited following 24 h treatment with 1 nM dasatinib (Figure 3d) and 10 nM Saracatinib (Figure 3e). Of note, phospho-Src Y416 was often not detected at basal levels in the MFD-1 cells using the same Western blot conditions. However, elevated levels of phospho-Src Y416 and total SRC were induced in the MFD-1 cells following 48 h CDDP treatment (Figure 3f). Therefore, this cell line was deemed relevant for further analysis.

### 3.5. Src Inhibition Synergizes with CDDP in OAC Cell Lines

Given the poor response rates in OAC patients that undergo neoadjuvant chemotherapy, we asked if Src inhibition could improve sensitivity to standard chemotherapy. CDDP is the main focus of this study as platinum-based chemotherapy is the main active DNA damaging agent used to treat OAC.

Dasatinib/CDDP combination experiments were assessed by MTT assays following 72 h ~IC_30(72 h)_ treatment (Appendix A). Dasatinib significantly reduced cell viability in OE33 (*p* < 0.001), FLO-1 (*p* < 0.01), and MFD1 (*p* < 0.05) cells when combined with CDDP (Figure 4a). To identify drug interactions, viability was assessed by MTT assays following 72 h treatment with three different doses of each drug in combination, and this was used to subsequently generate combination index (CI) values using the Chou and Talalay method [8]. CI is used to assess whether drug interaction, in terms of cell viability reduction, was due to additive (CI = 1) or synergistic (CI < 1) effects. From this, the CI values showed dasatinib/CDDP to be moderately synergistic in the OE33 and MFD-1 cell lines, with most CI values ranging between 0.4 and 0.8, and mainly additive or weakly synergistic (CI ranging between 0.8 and 1) in the FLO-1 cells (Figure 4b).

To assess if this synergy extended to other Src inhibitors, these experiments were repeated with saracatinib/CDDP combinations and showed that combination treatment further reduced OE33 (*p* < 0.01), FLO-1 (*p* < 0.05), and MFD-1 (*p* < 0.05) cell viability (Appendix A), in agreement with the dasatinib findings. Saracatinib also moderately synergised with CDDP in the OE33 cells (CI values ranging between 0.4 and 0.8), weakly synergised in the FLO-1 cells (CI values ranging between 0.8 and <1), and was mainly additive (CI = 1) in the MFD-1 cells (Appendix A). These results show that Src inhibitors can synergistically enhance CDDP treatment in multiple OAC cell line settings.

To determine the anti-tumour mechanisms of dasatinib/CDDP synergy in OAC cells, levels of cytoxicity and cell proliferation were investigated. PARP is a substrate for several proteases involved in cell death programs and results in proteolytic cleavage and loss of full-length PARP [21]. Western blot analysis of PARP was assessed following combination 72 h treatment of OE33, FLO-1, and MFD-1 cell with ~IC_30(72 h)_ doses of CDDP and dasatinib (Figure 4c). Combination treatment resulted in further loss of full-length PARP and/or an increase in cleaved PARP in all three cell lines, indicating elevated programmed cell death compared with either treatment alone. Cell cycle analysis, using propidium iodide flow cytometry, following 72 h combination treatment showed elevated subG1/G0 in the OE33 cells (*p* < 0.05) compared with either treatment alone, confirming an increase in cell death in this cell line (Figure 4d). However, no significant increase in subG1/G0 was observed in the FLO-1 cells, and although subG1/G0 was elevated in the MFD-1 following combination treatment, this was not significant.

Cell proliferation assays showed that, following 72 h dasatinib/CDDP combination treatment, the proliferation rate was significantly lower in OE33 (*p* < 0.01), FLO-1 (*p* < 0.05), and MFD-1 (*p* < 0.05) cells compared with either treatment alone (Figure 4e). Taken together, the results show that Src inhibition combined with CDDP resulted in reduced cell viability through increased cell death and/or abrogated proliferation.

### 3.6. Src Inhibition Synergizes with 5-FU in OAC Cell Lines

To confirm that targeting Src did not cause unintended resistance to other agents that are combined with CDDP in the clinic, the effect on another important constituent of neoadjuvant chemotherapy, 5-FU, was also briefly analysed.

Dasatinib was combined with 5-FU at ~IC_30(72 h)_ doses for 72 h (Appendix A) and the effect on OAC cell viability was assessed (Appendix A). Combination treatment significantly reduced the viability of OE33 and MFD1 cells compared with either treatment alone (*p* < 0.05). No significant change in FLO-1 cell viability was observed following combination treatment. To identify dasatinib/5-FU drug interactions, CI values were generated as before. From this, the CI values showed moderate synergy of dasatinib/5-FU in the OE33 and MFD-1 cell lines, with most CI values ranging between 0.4 and 0.8, and antagonistic (CI > 1) in the FLO-1 cells (Appendix A). Again, the study was extended to assess Src inhibitor saracatinib and showed that saracatinib/5-FU combination treatment significantly reduced OE33 (*p* < 0.01), FLO-1 (*p* < 0.05), and MFD-1 (*p* < 0.05) cell viability compared with either treatment alone (Appendix A). This was in agreement with the dasatinib/5-FU combination findings for the OE33 and MFD-1 cells. Moreover, saracatinib but not dasatinib enhanced 5-FU in the FLO-1 cells. Saracatinib also strongly synergised with 5-FU in the OE33 cells (CI < 0.4) and weakly synergised in the MFD-1 cells (CI values ranging between 0.8 and <1), and the CI appeared to be dose-dependent in the FLO-1 cells (Appendix A). These results show that Src inhibitors can also synergistically enhance 5-FU treatment in multiple OAC cell line settings.

### 3.7. Targeting Src Re-Sensitizes CDDP-Resistant Cells

An OE33 CDDP-resistant sub-line was generated by drugging OE33 cells continuously with incremental increased doses of CDDP over ~6 months. The OE33 sub-line was found to have an ~IC_30(72 h)_ dose of 4 µM CDDP compared with 1 µM in the original OE33 (OE33 parental) cell line (Appendix A, Appendix A). Given this fourfold increase in ~IC_30(72 h)_, the sub-line was labelled OE33 CDDP-resistant (CDDP-R) and, together with the OE33 parentals, this cell line pair was used to investigate the effect of Src inhibition on acquired CDDP resistance.

Western blot analysis was used to assess basal Src levels, and this showed that OE33 CDDP-R had higher levels of Src compared with their parental counterpart (Figure 5a), indicating Src as a potential mediator of acquired CDDP resistance. If true, Src inhibition should sensitise CDDP-R cells to CDDP. Therefore, dasatinib was combined with the parental ~IC_30(72 h)_ dose of 1 µM CDDP and cell viability was measured by MTT following 72 h treatment. Indeed, this combination treatment gave a significant reduction (*p* < 0.05) in OE33 CDDP-R cell viability to <70%, thus restoring CDDP sensitivity to that of the OE33 parental cells (Figure 5b). Furthermore, combining dasatinib with the OE33 CDDP-R ~IC_30(72 h)_ dose of CDDP resulted in additional loss of OE33 CDDP-R cell viability (Figure 5c).

### 3.8. Src Expression Is Elevated in Gastric Adenocarcinoma

To determine the levels of Src expression across multiple cancer types, data from The Human Protein Atlas [22] were assessed. The Cancer Genome Atlas (TCGA) RNA-seq and tumour section protein data were available for gastric adenocarcinoma (GAC), but no data were available for OAC. Clinically, oesophago-gastric adenocarcinomas are often grouped together because OACs strongly resemble the chromosomally unstable variant of GAC, suggestive of a single disease entity [23]. Furthermore, Barrett’s oesophagus, the pre-malignancy for OAC, has recently been shown to originate from gastric cardia [24]. TCGA RNA-seq data showed GAC to have the second highest median Src gene expression level across 17 cancer types (Appendix A) and the joint second highest levels of Src protein expression across 16 cancer types (Appendix A). A cohort of GAC tissue sections from 21 patients, stained for Src protein, showed only 2 negatively stained tissue sections (Appendix A). The remaining tissues sections showed 6 low, 9 medium, and 4 high levels of staining for Src protein expression.

## 4. Discussion

The primary aim of this study was to identify genes and pathways governing resistance to platinum-based neoadjuvant therapy, and subsequently target them to improve chemo-sensitivity. GSEA was used to identify genes and pathways that were differentially regulated between responders and non-responders, and thus potential mediators of resistance. The resulting data were assessed to select the 80 best candidate genes to take forward. Inhibition by siRNA was used to screen genes that, when inhibited, decreased cell viability and enhanced chemotherapy. Further validation using additional siRNAs and a panel of OAC cell lines identified seven genes (BACH1, F2RL2, FGF10, NNT, PKM2, SRC, and STAR) that consistently caused decreased cell viability alone or when combined with cisplatin or 5-FU. In parallel, a drug screen was performed to identify compounds that decrease OAC cell viability, focusing on compounds that had increased efficacy when combined with cisplatin. This parallel screen approach identified Src and several of its interactants as a potential pathway mediating platinum sensitivity in OAC. Further validation showed that Src inhibitors synergised with CDDP and 5-FU in multiple OAC settings in vitro. These results suggest that clinical targeting of Src in OAC could improve outcomes to neoadjuvant chemotherapy treatment.

Src is a non-receptor tyrosine kinase and a central nodal protein within multiple signalling pathways. It has a key role in driving tumourigenesis through mediating tumour survival, proliferation, angiogenesis, and invasion [17]. Indeed, Src has been investigated as a target in haematological malignancies, but it has only recently been considered a target in solid cancers. Aberrant signalling and/or overexpression is reported in many solid tumours such as breast, colorectal, lung, and oesophageal tumours [25]. 

Increased Src kinase activity has been reported in endoscopic tissue samples of Barrett’s oesophagus, the premalignant precursor of OAC (3–4-fold) and OAC (6-fold), compared with normal oesophagus control samples [26]. Src phosphorylates the tumour suppressor p27, inhibiting its regulatory function and subsequently leading to increased cell growth and proliferation. Barrett’s biopsies with high grade dysplasia displayed elevated Src kinase activity together with deregulated p27. Treatment of Barrett’s cell lines with dasatinib resulted in decreased Src activity, decreased p27 phosphorylation, and increased p27 protein levels. The dasatinib treated cells also had decreased proliferation and increased apoptosis [27]. These studies lend support to an important role for Src in the biology and progression of OAC and provide rationale for Src as a therapeutic target in this setting. However, targeting Src in OAC has thus far been unsuccessful. A Phase II clinical trial of saracatinib in metastatic and locally advanced gastro-oesophageal adenocarcinoma showed no objective response and thus insufficient activity as a single agent [28]. While Src inhibition alone has proven ineffective in OAC, there remains potential to combine with other therapies for an improved response. Interestingly, an in vitro OAC study has shown that Src hyperactivation confers resistance to the EGFR/HER2 inhibitor lapatinib and, subsequently, lapatanib resistant cells could be re-sensitised by combining lapatinib with saracatinib [29]. This would suggest a possible role for combined Src and HER2 inhibition in this OAC tumour subset.

Our study has several strengths compared with previous attempts to identify mediators of chemo-resistance to neoadjvant chemotherapy in OAC. Firstly, the large gene expression dataset of pre-chemotherapy biopsies with comprehensive clinicopathological information used in this study provides a high-powered discovery cohort. Secondly, the extensively validated parallel screen approach lends robustness to the finding of Src as a potential therapeutic target. Finally, the in vitro validation of Src inhibiton in parental and drug-resistant cell lines provides a strong rationale for further study of this target.

There are, however, limitations to our study. These include the use of a single siRNA and a single cell line to select potential candidate genes in the primary siRNA screen. Moreover, as the candidate gene list was limited to 80 genes, this likely overlooked other potential, novel targets. In addition, future work should include in vivo validation of the in vitro findings from the siRNA screen and the compound screen. During this study, the standard of care for neoadjuvant chemotherapy for resectable gastric or gastro-oesophageal junction adenocarcinoma in the United Kingdom changed from epirubicin, CDDP, and 5-FU/capecitabine (ECF/ECX) to a combination of 5-FU, leucovorin, oxaliplatin, and docetaxel (FLOT). While FLOT is still platinum-based, future work should include taxane-based chemotherapy in novel target selection. Finally, the selected OAC cell line panel used in this study may not be fully representative of this highly heterogeneous disease, thus further validation in a broader panel may be required.

In summary, our study suggests that Src inhibitors could enhance the OAC response to cisplatin and 5-FU. Further investigation using in vivo or near-patient models is required to validate the current findings followed by a re-appraisal of the efficacy of Src inhibition as a clinical strategy in OAC. Development of a companion biomarker may also be necessary to select sensitive patients for this combination treatment approach in clinical trials.

## 5. Conclusions

In conclusion, this functional genomic screening approach identified Src as a novel chemosensitization target in OAC with a parallel FDA compound screen also identifying the Src inhibitor, dasatinib. Further functional analysis showed the ability of Src inhibitors to synergise with, and overcome resistance to, standard OAC chemotherapies. This lends support to a combined therapeutic strategy of chemotherapy and Src inhibition to improve the current poor response rates to neoadjuvant chemotherapy in OAC.

## Figures and Tables

**Figure 1 cancers-14-03726-f001:**
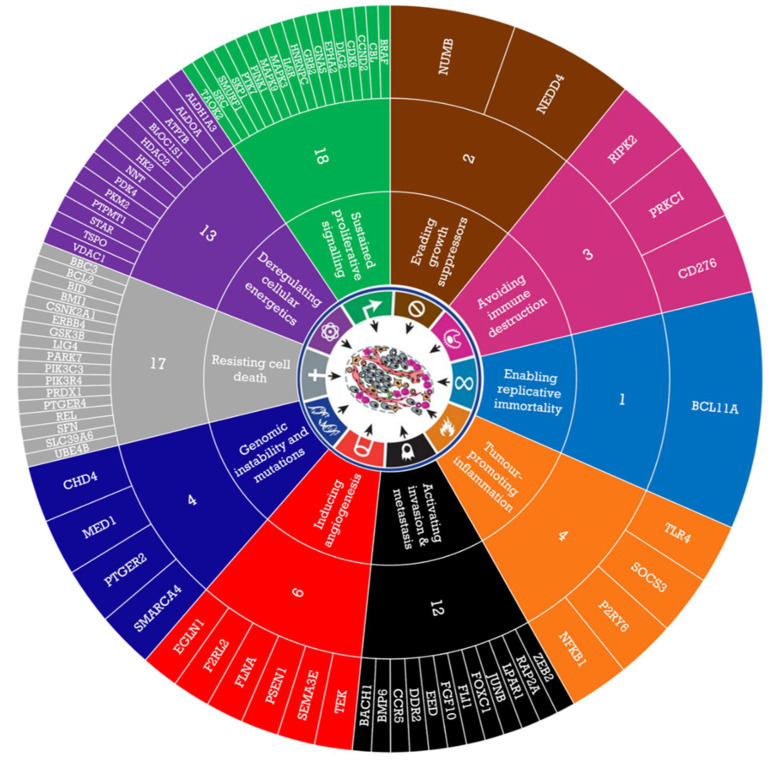
Candidate genes for siRNA screen. A total of 80 out of 784 enriched genes identified by GSEA were chosen for further evaluation based on the criteria (1) biological relevance to OAC, (2) expression in oesophageal/gastric tissue, (3) interacting genes and networks, and (4) targetability, and these can be further classified by Hanahan and Weinberg’s Hallmarks of Cancer.

**Figure 2 cancers-14-03726-f002:**
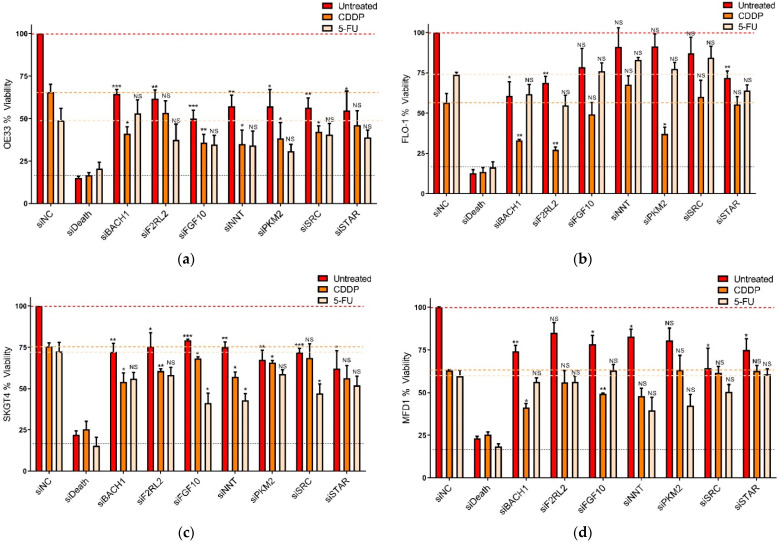
Positive hits identified by siRNA screening and the FDA-approved compound screen. (**a**) Positive hits from secondary siRNA screen were tested in (**a**) OE33, (**b**) FLO-1, (**c**) SKGT4, and (**d**) MFD1 cells. Cells were transfected with 10 nM siRNA for 24 h prior to 48 h treatment with ~IC_30(72 h)_ doses of CDDP or 5-FU. Viability was assessed by MTT. The results represent the mean ± SEM of triplicate experiments. Significance was assessed by Student’s *t*-test (* *p* < 0.05, ** *p* < 0.01, *** *p* < 0.001, NS, not significant). siNC, negative siRNA control; siDeath, positive siRNA control. (**e**) Drug class wheel representing 97 compounds that reduced OE33 or FLO-1 viability by ≥80%. Cells were treated with 1 µM of each library compound and viability was assessed by CellTiterGlo^®^ at 72 h.

**Figure 3 cancers-14-03726-f003:**
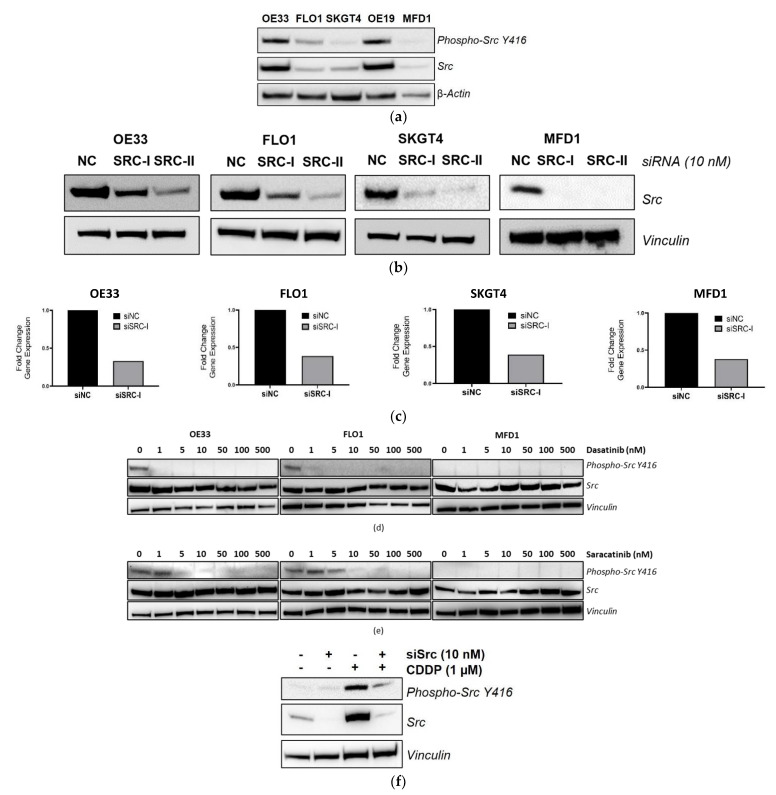
Src knockdown and inhibition in OAC cells. (**a**) Western blot analysis of basal phospho-Src Y416 and total Src protein in OAC cells with β-actin as a loading control. Src knockdown with 10 nM SRC-targeting siRNAs (SRC-I and SRC-II) for 72 h validated by (**b**) Western blot of protein levels with vinculin as a loading control and (**c**) Q-PCR of gene expression levels. NC, negative control siRNA. Western blot analysis of phospho-Src Y416 following 24 h treatment with (**d**) dasatinib or (**e**) saracatinib in OE33, FLO-1, and MFD1 cell lines with vinculin as a loading control. (**f**) Western blot analysis of phospho-Src Y416 and total Src in MFD-1 cells transfected with (**+**) 10 nM siSRC or (**−**) 10 nM siNC for 24 h prior to 48 h CDDP (1 µM) treatment with vinculin as a loading control.

**Figure 4 cancers-14-03726-f004:**
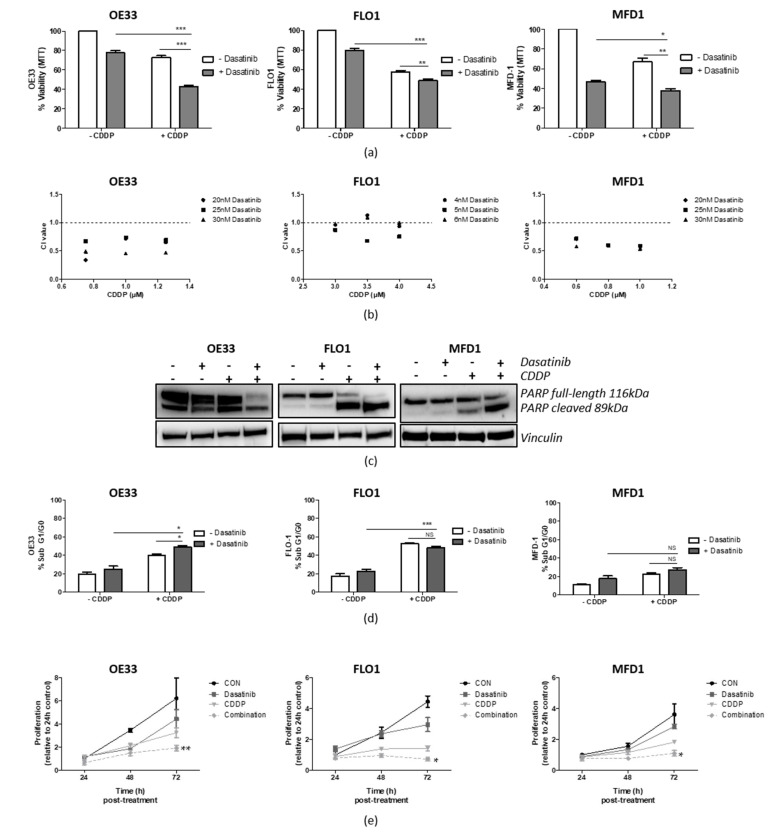
Dasatinib synergises with CDDP in OAC cell lines: (**a**) OAC cells were treated with ~IC_30__(72 h)_ doses of dasatinib/CDDP for 72 h and viability was assessed by MTT. The results represent the mean ± SEM of triplicate experiments. (**b**) Drug interactions were then assessed following 72 h dasatinib/CDDP combinations (≤~IC_30__(72 h)_) by combination index (CI), where a CI < 1 indicates synergy, CI = 1 indicates additive, and CI > 1 indicates an antagonistic interaction. (**c**) Western blot analysis of PARP in OE33 FLO-1 and MFD-1 cells following 72 h ~IC_30__(72 h)_ dasatinib/CDDP combinations with vinculin as a loading control. (**d**) Propidium iodide flow cytometry of Sub-G1 phase following 72 h ~IC_30__(72 h)_ CDDP/dasatinib combination. (**e**) Proliferation rates at 24, 48, and 72 h post combination treatment. The results represent the mean ± SEM of triplicate experiments. Significance was assessed using Student’s *t*-test (* *p* < 0.05, ** *p* < 0.01, *** *p* < 0.001). CON, control; NS, not significant.

**Figure 5 cancers-14-03726-f005:**
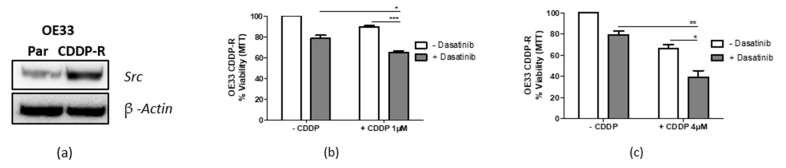
Targeting Src re-sensitizes CDDP-resistant OE33 cells. (**a**) OE33 parental (Par) and OE33 CDDP-resistant (CDDP-R) were assessed for SRC levels by Western blot and β-actin was used as a loading control. (**b**) OE33 CDDP-R cells were treated with dasatinib (~IC_30 (72 h)_) in combination with 1 µM CDDP (OE33 parental ~IC_30(72 h)_) or (**c**) 4 µM CDDP (OE33 CDDP-R ~IC_30(72 h)_) for 72 h. Viability was assessed using MTT assays and Student’s *t*-test was applied to test for significance with * *p* < 0.05, ** *p* < 0.01, *** *p* < 0.001, and ns (non-significant). The results represent the mean of triplicate experiments ± SEM.

**Table 1 cancers-14-03726-t001:** List of enriched pathways (*p* < 0.01) identified by GSEA from non-responder and responder pre-treatment tumour biopsies.

	GSEA Pathway Names
Non- Responders (C2)	1 AMIT_EGF_RESPONSE_120_MCF10 2 BIOCARTA_RHO_PATHWAY 3 REACTOME_METAL_ION_PC12_SLC_TRANSPORTERS
Non- Responders (C5)	1 GO_ENERGY_COUPLED_PROTON_TRANSPORT_DOWN_ELECTROCHEMICAL_GRADIENT 2 GO_ATP_BIOSYNTHETIC_PROCESS 3 GO_PCG_PROTEIN_COMPLEX 4 GO_MITOCHONDRIAL_ATP_SYNTHESIS_COUPLED_PROTON_TRANSPORT 5 GO_OXIDOREDUCTASE_ACTIVITY_OXIDISING_METAL_IONS 6 GO_POSPHATIDYL_ACYL_CHAIN_REMODELLING 7 GO_MITOCHONDRIAL_MEMBRANE_PART 8 GO_ORGANELLAR_LARGE_RIBOSOMAL_SUBUNIT 9 GO_INTRINSIC_COMPONENT_OF_MITOCHONDRIAL_MEMBRANE 10 GO_POSITIVE_REGULATION_OF_RESPONSE_TO_EXTRACELLULAR_STIMULUS 11 GO_MITOCHONDRIAL_TRANSPORT 12 GO_PROTEIN_TARGETING_TO_MEMBRANE 13 GO_MULTICELLULAR_ORGANISMAL_HOMEOSTASIS 14 GO_VACUOLAR_TRANSPORT 15 GO_MACROAUTOPHAGY
Responders (C2)	1 PID_IL2_STAT5_PATHWAY 2 REACTOME_ANTIGEN_ACTIVATES_B_CELL RECEPTOR 3 PLASARI_TGFB1_SIGNALLING_VIA_NFIC_1HR_UP 4 REACTOME_AMINO_ACID_TRANSPORT_ACROSS_THE_PLASMA_MEMBRANE 5 BIOCARTA_DC_PATHWAY 6 BANDRES_RESPONSE_TO_CARMUSTIN_MGMT_48HR_UP 7 BROWNE_HCMV_INFECTION_30MIN_UP 8 REACTOME_IL_3_5_AND_GM_SCF_SIGNALLING
Responders (C5)	1 GO_PHOSPHOLIPASE_C_ACTIVITY 2 GO_CELLULAR_RESPONSE_TO_PROSTAGLANDIN_STIMULUS 3 GO_ALDEHYDE_DEHYDROGENASE_NAD_ACTIVITY 4 GO_REGULATION_OF_FIBROBLAST_MIGRATION 5 GO_RESPONSE_TO_PROSTAGLANDIN 6 GO_HEART_TRABECULA_MORPHOGENESIS 7 GO_CELLULAR_RESPONSE_TO_PROSTAGLANDIN_E_STIMULUS 8 GO_CELLUALAR_TO LITHIUM_ION 9 GO_RESPONSE_TO_PROSTAGLANDIN_E 10 GO_REGULATION_OF_INTERFERON_GAMMA_BIOSYNTHETIC_PROCESS 11 GO_DENDRITIC_SHAFT 12 GO_RNA_POLYMERASE_II_DISTAL_ENHANCER_SEQUENCE_SPECIFIC_BINDING 13 GO_NEUROMUSCULAR_JUNCTION_DEVELOPMENT 14 GO_TRANSCRIPTIONAL_ACTIVATOR_ACTIVITY_RNA_POLYMERASE_II_DISTAL_ENHANCER_SEQUENCE_SPECIFIC_BINDING 15 GO_REGULATION_OF_EXTENT_OF_CELL_GROWTH 16 GO_ESTABLISHMENT_OR_MAINTENANCE_OF_CELL_POLARITY 17 GO_BASOLATERAL_PLASMA_MEMBRANE

## Data Availability

Raw expression data are available at the Array Express repository (Accession Number E-MTAB-6969).

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
