# Peer review of "Identification of Src as a Therapeutic Target in Oesophageal Adenocarcinoma through Functional Genomic and High-Throughput Drug Screening Approaches"

_cancers, 2022, doi:10.3390/cancers14153726_

Round 1

Reviewer 1 Report

This manuscript using bioinformatics analysis of transcriptional data and functional siRNA screen identified 8 genes that showed significant tumor suppression after siRNA knockdown. Among the 8 genes, the author selected SRC for further investigation. They found that SRC inhibition using specific inhibitors synergized with CDDP or 5-FU and sensitized CDDP resistant cells to CDDP treatment. Generally, the manuscript was well written and organized and provided interesting information for possible application of SRC inhibitors in OAC treatment. However, the data provided were all from in vitro cell models, lack in vivo experiments and human tissue data that significantly compromised the value of the manuscript. Some minor points also need to be addressed before accepting for publication.

11) In vivo xenografting mice model or PDX model is needed to validate the in vitro results presented in this manuscript; using CDDP/5-FU, Dasatinib or combination.

2) Figure 3. Only 2 of 5 OAC cell lines showed overexpression of SRC. What is the SRC expression in human OAC tissue? It is suggested to add IHC staining of at least 50 OAC tissues.

3)      Figure 3 panels (d) and (e), the loading control of Vincilin was not even, need to re-do western blot to normalize the loading control.

4)      Figure 3 panel (f) only showed the results in MFD1 cells. What are the changes of p-SRC and SRC after CDDP in other cell lines? Need to show at least two cell lines.

5)      Figure 4, (b), the synergy analysis. There is an online tool to analyze drug synergy: https://synergyfinderplus.org. Please use the tool to add the representative synergy score and map in manuscript.

6)      Figure 4, (c), the western blot results did not match the observed results in panel (a). For example, in OE33 cells, Dasatinib and combination showed lower levels of c-PARP even compared to control? In FLO1 and MFD1 cells, Dasatinib did not show c-PARP while it displayed significant reduction of tumor viability in (a).

7)      Figure 4, (d), the results also not reasonable, for example, in MDF1 cells, there was a significant reduction of cell viability in combination treatment in (a) and also marked induction of apoptosis as showed in (c), but it did not show any significant change of G0/G1. It is suggested to use annexin V flow cytometry to monitor apoptosis. It is also possible that other type of cell death, in addition to apoptosis, occurred.

8)      Figure 4, (e), what assay used for proliferation rate measurement? It is suggested to use BrdU incorporation assay, not just count the cell number.

9)      Figure 5, missing figure legend. (a) needs to show p-SRC.  For (b) and (c), need to show the dose-response curve and IC50 change, generated by Prism software, described in Method section.

10)   The labeling in Supplementary fig.1 a is confusing. What is the difference between siGENE and siRNA? Please add p-values. In supplementary fig. 1b, please add synergy score from synergyfinderplus.org.

Page 11, line 351-352, the font looks different. 

Author Response

We would like to thank the reviewers for their comments. The following revisions have been made in response to those comments.

Responses to reviewer 1 comments:

  1. Due to financial and time constraints, this paper does not contain in vivo However, we feel that the extensive discovery and in vitro elements of the study provide enough evidence to warrant in vivo investigations as a future direction. This is now commented on within the discussion. Despite the lack of in vivo data, we feel the clinical basis of the study, taken from a large patient cohort, is a major strength.
  2. The reviewer suggests including IHC staining of 50 OAC tissues for Src expression. However, this would not be possible within the 10-day timeframe for resubmission. This study does however, use clinical data to select SRC for RNAi screening as it had enriched gene expression in responding patients vs. non-responding patients.

Publicly available gastric adenocarcinoma data, including IHC stained tissue sections, for Src gene and protein expression has been taken from The Human protein atlas. There is now an additional results section to cover this (Section 3.8, Fig 5a, b, c).

  1. The reviewer asks for vinculin blots to be re-done in Fig 3e and 3f due to uneven loading. The purpose of these figures is to show that Src inhibitors dasatinib and saracatinib inhibits phospho-Src Y416 in OAC cell lines at low nanomolar concentrations. We do not feel that the slight differences in loading at higher concentrations takes away from this conclusion and this is supported by the normalized densitometry data that is now included in the supplementary material.
  2. The reviewer asks for inducible phospho-Src to be shown in at least 2 cell lines in Fig 3f. We do not feel this is necessary as the purpose of Fig. 3e and 3f is to provide a rationale for using compounds that inhibit phospho-Src in the 3 OAC cell lines. For this we need to show the presence of phospho-Src, either basal or inducible. We therefore show that phospho-Src is present basally in the OE33 and FLO-1 cells but not the MFD-1 cell lines but go onto show that phospho-Src can be induced in MFD-1 by cisplatin. Thus, providing a rationale for using dasatinib and saracatinib in all 3 OAC cell lines.
  3. The reviewer suggests performing synergy analysis using synergyfinderplus.org. While this is an acceptable form of analysing drug synergy, our study also applies a widely cited (3188 via Web of Science) and acceptable form of analysing drug synergy using CompuSyn software to calculate combination index (Chou, Cancer Res 2010).
  4. The reviewer states that that western blot cleaved PARP (cytotoxic) results for dasatinib treatment alone in Fig. 4c do not reflect what is observed for the cell viability data in Fig 4a. This is a reasonable observation as reduced cell viability can be attributable to cytotoxic and/or cytostatic effects. In this instance, dasatinib does not appear to induce a cytotoxic effect as there are no changes in PARP. Therefore, dasatinib-induced decreases in cell viability are mainly the result of a cytostatic effect as shown by reduced proliferation rates in Fig 4e.

The reviewer also states that less cleaved PARP is observed for combination treatment (4c) in the OE33 and does not reflect the reduced viability data (4a). However, for the OE33, the combination treatment resulted in loss of full-length PARP compared to either treatment alone (4c), as well as reduced proliferation rates (4e) and we feel that this does reflect the reduction in cell viability observed in 4a.

  1. The reviewer rightly points out that while increased cleaved PARP (cytotoxicity) is observed following combination treatment in the MFD1 (4c), we did not observe a significant difference in subG1/G0 (another indicator of cytotoxicity). However, the subG1/G0 assay is not definitive for cytotoxicity. A more appropriate assay to confirm cytotoxicity would be Annexin V and this is suggested by the reviewer. We now address the issue relating to the sub G1/G0 assay within the results section and state that further mechanistic analysis is warranted. However, it would not be possible to perform triplicate annexin V assays within a 10-day time frame.
  2. The reviewer suggests using BrdU incorporation to measure cell proliferation. For this study we measured viable cell counts taken over 24, 48 and 72 h and this is now included in the materials and methods section. While there is a vast array of acceptable methods to measure cell proliferation, we feel that this is a widely used and acceptable measure of proliferation rates.
  3. Missing legend in Fig 5 is now added. Dose response curve is now added to supplementary material with changes in ~IC30(72h).
  4. The confusing labelling in supplementary Fig.1a has now been removed and p-values are represented on each graph. For supplementary Fig. 1b, this was not a synergy experiment but a screen using multiple siRNA sequences to rule out off-target effects. Therefore, synergy scores are not warranted.
  5. Page 11, line 351-353: the font has now been adjusted.

Reviewer 2 Report

The manuscript reports Src as a potential target for combination treatment of esophageal adenocarcinoma with cisplatin. Src was identified as a target through bioinformatic analysis with GSEA using previously reported gene expression data of esophageal cancer patients followed by siRNA screening and, independently through drug screening. The authors demonstrated that knockdown of Src expression by siRNA or inhibition of Src phosphorylation resulted in synergy or additive effect in combined treatment with cisplatin. Although lack of in vivo analysis is a main weak point, the authors provided enough in vitro evidences in this report that could warrant subsequent in vivo analysis.

Here are a few comments that are to be addressed by the authors.

1. Axis labels and numbers in graphs in Fig. 2 and Fig 3 are not readily legible. Need to improve the quality of the figures.

2. ‘Supplementary Table 3’ (line 285) is not found.

3. In line 168, ‘at false discovery rate (FDR) p<0.25’ does not reconcile with the description at lines 203-205.

4. There are at least two previous reports on co-treatment of dasatinib and cisplatin to esophageal cancer cells which should be included in the ‘Discussion’.

Wen P, Dayyani F, Tao R, Zhong X. Ann Transl Med. 2022 Jan;10(2):70

Chen J, Lan T, Zhang W, Dong L, Kang N, Fu M, Liu B, Liu K, Zhang C, Hou J, Zhan Q. Arch Biochem Biophys. 2015 Jun 1;575:38-45.

5. There are a few typographical errors that should be corrected. Especially, a space is needed between number and unit.

Author Response

We would like to thank the reviewers for their comments. The following revisions have been made in response to those comments.

Responses to reviewer 2 comments:

  1. The quality and legibility of axis labels and numbers in graphs in Fig. 2 and Fig. 3 have now been improved.
  2. Supplementary Table 3 now included.
  3. The typo of False discovery rate (FDR) p<0.25 (line 168) has now been addressed. This has been edited to state a nominal p value of 0.1 and an FDR of 0.25. This now reconciles with what is stated in lines 203-205.
  4. The reviewer kindly suggests referencing 2 previous reports of cisplatin and dasatinib co-treatment in oesophageal cancer. These were Wen et al., 2022 and Chen et al., 2015. However, both of these studies looked at oesophageal squamous cell carcinoma. Our study aims to specifically addresses oesophageal adenocarcinoma. We therefore do not feel it is appropriate to reference these studies.
  5. The typographical errors, particularly spaces between numbers and units, have now been addressed throughout.

Reviewer 3 Report

Authors have identified a new possible route to enhance chemotherapy sensitivity in Oesophageal Adenocarcinoma. Authors demonstrated that combining Src inhibitors synergize with common-used platinum base compounds in OAC treatment, significantly enhancing the anti-cancer effects of both strategies alone. Src was selected after several layers of experiments performed on a battery of eighty initial genes. These experiments englobed Gene Set Enrichment Analysis, literature review and knock-down experiments. Authors proved dasatinib and saracatinib synergize with CDDP and 5-FU to reduce the viability of OAC cell lines. Importantly, CDDP resistant cell lines showed sensitivity to low levels CDDP again when combined with Src inhibitors. In addition, Src was identified as overexpressed in CDDP resistant lines.

Honestly, I found the paper well structured, well written, and without significant flaws. The message is clear, and the results are readily usable by other groups who wish to continue investigating these lines.

I have minor comments I would like the authors to address:

1) I suggest authors add a high-quality version of Figure 1 to the supplementary files as they have done for the western blot images. Likewise, I suggest adding a high-resolution image of the drug wheel of Fig 2e) to the Supp images. The current resolution is unreadable.

2) Supplementary Figure 1) I understand the label 'Untx' means untreated as it is referenced in some other legends. All should read 'untreated'.

3) I suggest authors note with a start (*) in supp. figure 1a the genes selected to supp figure 1b, so that readers can quickly assess the results for those genes.

4) In supp figure 1a there's the label 'siRNA'. Did the authors mean "siGENE"?

5) It would be valuable to identify the seven genes after the secondary siRNA screen in figure 1. Maybe with a start (*) as outer layer.

6) Figure 5 does not have a caption.

7) Supplementary table 3 was not uploaded.

8) Slightly increase labels for y axis, x axis, and legend in figure 2.

9)

# Figure 3

- panel b) why FLO1 NC intensity is very high, and panel a) Src and Phospho-Src Y416 is so low? Shouldn't these be the same? MFD1 also does not show a band in panel a) and does show a band in panel b). In general terms, what is the expression variability in these assays when observed by western blot?

- increase labels in panel c): y label, x label, yticks, xticks, and legend.

10)

# Figure 4

- instead of using "CON" as a label, use the entire word or explain the acronym in the figure legend.

Other typos: 

line 28 - "Abstract: Drug-resistance limits the effectiveness of OAC" - write the full OAC name here.

line 54: "[2],[3]" -> "[2, 3]"

line 151 - "chemotherapy resgime" -> did authors mean "chemotherapy regimen"?

line 291 - "[16],[17]"

line 443 - "[20],[21]" to "[20, 21]"

line 473 - "During the study" to "During this study,"

line 490 - "support to a combination therapeutic" correct to "support to a combined therapeutic"

Sup fig 1 - "10nM siRNA for 24 followed" -> 24h

Both instances of "neoadjuvant" and "neo-adjuvant" appear in the text. Select one of the forms throughout the text. 

Author Response

We would like to thank the reviewers for their comments. The following revisions have been made in response to those comments.

Responses to reviewer 3 comments:

  1. Higher quality and more legible versions of Fig. 1 and 2e have now been added to supplementary material.
  2. Supplementary Fig. 1 “Untx” has been changed to “untreated”.
  3. Positive hits from Supplementary Fig1a carried forward to Supplementary Fig 1b are now highlighted with #.
  4. The confusing labelling in supplementary Fig.1a has now been removed.
  5. Positive hits from Supplementary Fig. 1b carried forward to Fig. 2a, 2b, 2c and 2d are now highlighted with #.
  6. 5 figure legend has now been added.
  7. Supplementary Table 3 is now included.
  8. Axis labels and legend for Fig. 2 have now been enlarged and the figure has been enhanced to improve legibility.
  9. Reviewer asks why band intensity for Src changes for NC FLO-1 from Fig. 3a to 3b. The reason for these differences is variable exposure time for each blot. Fig. 3a would have had a shorter exposure time for optimum image capture so as not to overexpose/saturate the higher levels of Src in OE33 and OE19 cells that appear on the same blot. For Fig 3b FLO1 blot, a longer exposure time would have been applied to obtain the optimum image for the lower levels of Src in FLO1 NC vs. knockdown. This is normal practice in Western blotting in order to obtain optimum image capture. That is why Western blotting can only be used for relative quantification. That being you can only compare intensity of bands to samples that appear on the same blot. You can never compare intensity of bands that appear on a separate blot, especially if the images have been captured separately.

Panel 3c has now been enhanced accordingly.

  1. CON abbreviation of control is now explained in the Fig. 4e figure legend.
  2. Typos on line 28, 151, 473, 490, supplementary fig.1 have now been addressed. Reference numbers in separate brackets have now been combined.

Round 2

Reviewer 1 Report

These experiments are important to improve the quality of the manuscript and support a solid conclusion. 

2)      Figure 3. Only 2 of 5 OAC cell lines showed overexpression of SRC. What is the SRC expression in human OAC tissue? It is suggested to add IHC staining of at least 50 OAC tissues.

3)      Figure 3 panels (d) and (e), the loading control of Vincilin was not even, need to re-do western blot to normalize the loading control.

4)      Figure 3 panel (f) only showed the results in MFD1 cells. What are the changes of p-SRC and SRC after CDDP in other cell lines? Need to show at least two cell lines.

5)      Figure 4, (b), the synergy analysis. There is an online tool to analyze drug synergy: https://synergyfinderplus.orgPlease use the tool to add the representative synergy score and map in manuscript.

6)      Figure 4, (c), the western blot results did not match the observed results in panel (a). For example, in OE33 cells, Dasatinib and combination showed lower levels of c-PARP even compared to control? In FLO1 and MFD1 cells, Dasatinib did not show c-PARP while it displayed significant reduction of tumor viability in (a).

7)      Figure 4, (d), the results also not reasonable, for example, in MDF1 cells, there was a significant reduction of cell viability in combination treatment in (a) and also marked induction of apoptosis as showed in (c), but it did not show any significant change of G0/G1. It is suggested to use annexin V flow cytometry to monitor apoptosis. It is also possible that other type of cell death, in addition to apoptosis, occurred.

8)      Figure 4, (e), what assay used for proliferation rate measurement? It is suggested to use BrdU incorporation assay, not just count the cell number.

Author Response

In response to the editor's request to discuss limitations of our study, please find the following paragraph included at lines 510-522. 

"There are, however, limitations to our study. These include the use of a single siRNA and a single cell line, to select potential candidate genes in the primary siRNA screen. Also, since the candidate gene list was limited to 80 genes, this likely overlooked other potential, novel targets. In addition, future work should include in vivo validation of the in vitro findings from the siRNA screen and the compound screen. During this study, the standard of care for neoadjuvant chemotherapy for resectable gastric or gastro-oesophageal junction adenocarcinoma in the UK changed from epirubicin, CDDP and 5-FU/capecitabine (ECF/ECX) to a combination of 5-FU, leucovorin, oxaliplatin and docetaxel (FLOT). Whilst FLOT is still platinum-based, future work should include taxane-based chemotherapy in novel target selection. Finally, the selected OAC cell line panel used in this study may not be fully representative of this highly heterogeneous disease and so further validation in a broader panel may be required."
